# A GPU-accelerated fuzzy method for real-time CT volume filtering

**Celia Tendero Delicado**[1]*, **Mónica Chillarón Pérez**[1], **Josep Arnal García**[2], **Vicent Vidal Gimeno**[1], **Esther Blanco Pérez**[3]

1 Department of Computer Systems and Computation, Universitat Politècnica de València, Valencia, Valencia, Spain, 2 Departamento de Ciencia de la Computación e Inteligencia Artificial, Universidad de Alicante, San Vicente del Raspeig, Alicante, Spain, 3 Department of Radiology, University Hospital de La Ribera, Alzira, Valencia, Spain

* ctendel@posgrado.upv.es

**Data Availability Statement:** The databases utilised in this work were compiled by the Mayo Clinic and were made publicly available in the Cancer Imaging Archive (https://www.

## Abstract

During acquisition and reconstruction, medical images may become noisy and lose diagnostic quality. In the case of CT scans, obtaining less noisy images results in a higher radiation dose being administered to the patient. Filtering techniques can be utilized to reduce radiation without losing diagnosis capabilities. The objective in this work is to obtain an implementation of a filter capable of processing medical images in real-time. To achieve this we have developed several filter methods based on fuzzy logic, and their GPU implementations, to reduce mixed Gaussian-impulsive noise. These filters have been developed to work in attenuation coefficients so as to not lose any information from the CT scans. The testing volumes come from the Mayo clinic database and consist of CT volumes at full and at simulated low dose. The GPU parallelizations reach speedups of over 2700 and take less than 0.1 seconds to filter more than 300 slices. In terms of quality the filter is competitive with other state of the art algorithmic and AI filters. The proposed method obtains good performance in terms of quality and the parallelization results in real-time filtering.

## Introduction

In medical imaging (e.g., X-rays, Magnetic Resonance Imaging (MRI), Computerized Tomography (CT)) the image quality greatly influences the diagnosis of a disease, and as such, filtering methods that detect and reduce noise become essential. CT has become an essential diagnostic tool in recent years, but the radiation necessary to generate acceptable images makes it a hazard for many patients, be it due to their age [1], their body mass index [2] or the frequency of tests [3]. Generating valid diagnostic images at a lower dose would reduce the risk of developing pathologies derived from radiation exposure for all patients. However, the reduction of radiation dose reduces in turn the amount of data gathered from the tomography and thus the images reconstructed at lower radiation doses have significant noise [4]. It is therefore necessary to denoise low dose CT images for correct image analysis.

The most common noise type in these kinds of images is the Gaussian noise, which is introduced during acquisition [5]. There exist many algorithms for filtering Gaussian noise [6–11].

cancerimagingarchive.net/collection/ldct-and-projection-data/). The code for the filter is available in the following repository: https://github.com/anicecloud/FGI_filter/.

**Funding:** This research has been supported by the TED project Grant Reference TED2021-131091B-I00 funded by MCIN/AEI/ 10.13039/501100011033 and by the "European Union NextGenerationEU/PRTR". Funding for open access charge: CRUE-Universitat Politècnica de València. The funders had no role in study design, data collection and analysis, decision to publish, or preparation of the manuscript.

**Competing interests:** The authors have declared that no competing interests exist.

Conventional filters like the bilateral filter [7] have been adapted for their use in medical images and present good results [12]. In Bhonsle et al. [13] the authors used the filter successfully on medical images corrupted with Gaussian noise, but encountered problems when filtering impulsive salt and pepper noise. As the noise present in CT images usually appears accompanied of other noise types there is a need for filtering techniques capable of reducing complex noise [14].

There are some recent approaches to complex noise reduction: in Zhang et al. [15] the authors study a filter for combined Poisson-Gaussian noise in fluorescent microscopy, in Kusnik and Smolka [16] the authors implement a Gaussian-impulsive noise mean shift filter for color images and in Li et al. [17] the authors study an image-to-image translation method for low-dose CT image denoising.

In addition, new artificial intelligence-based filtering methods are emerging, applied to both general and medical imaging. In particular, for low-dose CT images, many methods have been presented in recent years, based on CNNs (Convolutional Neural Networks) and GANs (Generative Adversarial Networks). For instance, Li et al. [18] propose a progressive cyclical convolutional neural network (PCCNN), a new unsupervised denoising framework using unpaired CT data. Patwari et al. [19] propose a denoising framework which comprises two deep CNNs trained via a Deep-Q reinforcement learning task used to tune the parameters of the denoising methods used on both the CT images and the sinograms. As for GANs, Huang et al. [20] propose DU-GAN, which leverages U-Net-based discriminators in the GAN framework to learn both global and local differences between the denoised and normal-dose images in both image and gradient domains. These are only a few examples of new AI-based denoising methods, but there are several extensive review papers on this issue [21, 22].

The method studied in this paper aims to reduce mixed Gaussian and impulsive noise, utilising fuzzy logic. This paper hypothesizes that a GPU-accelerated fuzzy filter can achieve both real-time processing and superior noise reduction in low-dose CT images compared to existing methods.

Unlike other imaging techniques like X-rays, acquisition in CT is done in volumes, which may consist of hundreds of different slices. The consequent performance cost of filtering multiple images makes most filters not applicable for real-time processing. Parallel computing becomes the most appropriate way to obtain real-time processing, specifically processing in graphics processing units (GPU) [23]. The filter studied in this paper is based on the fuzzy method described by Camarena et al. for RGB images [24], and it was adapted for its use with medical images by Arnal et al. [25]. The fuzzy Gaussian-impulsive (FGI) filter has shown good performance with mixed Gaussian-impulsive noise in CT images but is limited by the input format. Quality results are competitive, and the version for medical images has been parallelized for multi-core systems, but the performance is not sufficient for real-time processing of CT volumes. CT images are usually visualized in DICOM format, the Hounsfield Units (HU) utilized in this format are significant and indicate the density of the elements described [26]. The new implementation studied modifies the algorithm to permit GPU parallelization and changes data transfer to permit filtering full DICOM CT volumes stored in Hounsfield Units (HU) that have been converted to attenuation coefficients using the water attenuation coefficient. On top of that, several memory management improvements were implemented, resulting in four different versions of the algorithm. We have analyzed the presented parallel algorithms on GPU, achieving favorable speed-up outcomes, showing that the method presented is a viable and effective strategy for conducting real-time image processing. We have conducted a comparative analysis between the FGI method and other notable filters that have been successfully utilized in medical images: the Fuzzy Peer Group Averaging Filter (FPGA) [27, 28], the bilateral filter [7, 13], the WavResNet [29] and the CNCL [30]. And in terms of

speed-up, we have compared the GPU parallelization with the existing CPU parallelization [25].

In the experiments, we used CT volumes from the Mayo clinic database [31]. In particular, the volumes studied have a full dose and a reduced dose version and come from the DICOM-CT-PD database [32], which allowed us to evaluate the quality performance of the filters. The performance has been evaluated using *Peak Signal to Noise Ratio* (PSNR), and the *Structural Similarity Index* (SSIM) [6, 33].

This paper is organized in 5 sections. Section Materials and methods explains the proposed method, experimental results are analyzed in section Results, the discussion is presented in section Discussion and the conclusions are described in section Conclusion.

## Materials and methods

The filtering method proposed is based on fuzzy rules and designed to eliminate both Gaussian and impulsive noise, pixel by pixel. Given a gray-scale image $I$, and a filtering window $W$ of size $n \times n$ with central pixel $x$. The method computes a new value for $x$, $\hat{x}$ as a weighted average of the $q$ closest pixels from the window $W$ like

$$\hat{x} = \frac{\sum_{j=1}^{q} \omega_j \cdot x^j}{\sum_{j=1}^{q} \omega_j}. \tag{1}$$

Where the weights $\omega_j \in [0, 1]$ are determined following a fuzzy rule-based model.

### Computing impulsivity

In a first phase, each pixel in the image $x_i$ gets assigned an impulsivity degree $\delta(x_i)$. A $n \times n$ window $W$ with central pixel $x_i$ is reordered based in the distance of every $x^j$ to the centre, leaving $x_i$ first on the list. Then the $ROAD_r$ is calculated as the sum of the first $r$ distances [34]:

$$ROAD_r(x_i) = \sum_{j=0}^{r} d(x_i, x^j), \tag{2}$$

Low values of $ROAD_r$ indicate a central pixel that is similar to those in its window, while a high value of $ROAD_r$ indicate that the pixel is very different to those around it and thus we can consider it impulsive. We then define this degree of impulsivity for a pixel $x_i$ with the membership function

$$\delta(x_i) = \begin{cases} 0, & ROAD_r \leq p_1, \\ \dfrac{ROAD_r - p_1}{p_2 - p_1}, & p_1 < ROAD_r < p_2, \\ 1, & p_2 \leq ROAD_r. \end{cases} \tag{3}$$

The parameters $p_1$ and $p_2$ were determined previously for pixel values in the range [0, 255] [25]. For the version that works in volumes we have scaled them to equivalent values in attenuation coefficients.

### Computing the similarity degree

The similarity between the central pixel $x_i$ and the rest of the pixels in its window $W$ can take the values "high", "moderate" and "low". Being $L_1$ the distance with the central pixel $d(x_i, x^j)$,

the "high" similarity value is determined by the membership function:

$$\mu_H(x_i, x^j) = \begin{cases} 1, & L_1 \leq p_3, \\ \dfrac{-L_1}{3 * p_3} + \dfrac{4}{3}, & p_3 < L_1 < 4 * p_3, \\ 0, & 4 * p_3 \leq L_1. \end{cases} \tag{4}$$

And the "low" value, using fuzzy negation, is given by:

$$\mu_L(x_i, x^j) = 1 - \mu_H(x_i, x^j). \tag{5}$$

Lastly, the "moderate" similarity value is defined as:

$$\mu_M(x_i, x^j) = \begin{cases} \dfrac{L_1 - p_3}{p_3}, & p_3 < L_1 < 2 * p_3, \\ 1, & 2 * p_3 \leq L_1 \leq 3 * p_3, \\ \dfrac{4 * p_3 - L_1}{p_3}, & 3 * p_3 < L_1 < 4 * p_3, \\ 0, & elsewhere. \end{cases} \tag{6}$$

The $p_3$ utilized in the membership functions in [25] was a function of the Gaussian noise applied in the simulated noise images ($p_3 = (0.997 * \sigma + 1.96)$). Since we are using natural CT in attenuation coefficients, we have fixed the value as $p_3 = 1.96/255$. In Fig 1 we show the fuzzy sets that compose the similarity degree according to the distance $L_1$.

## Fuzzy rules

The value for the weight $\omega_j$ corresponding to to $x^j$ is obtained by defuzzification. To calculate the average weight in Eq (1), the following three rule fuzzy system is used:

Fuzzy Rule 1: IF ($x^j$ is not impulsive AND $x_i$ is impulsive AND $x_i$ and $x^j$ have moderate similarity) THEN $\omega_j$ is moderate

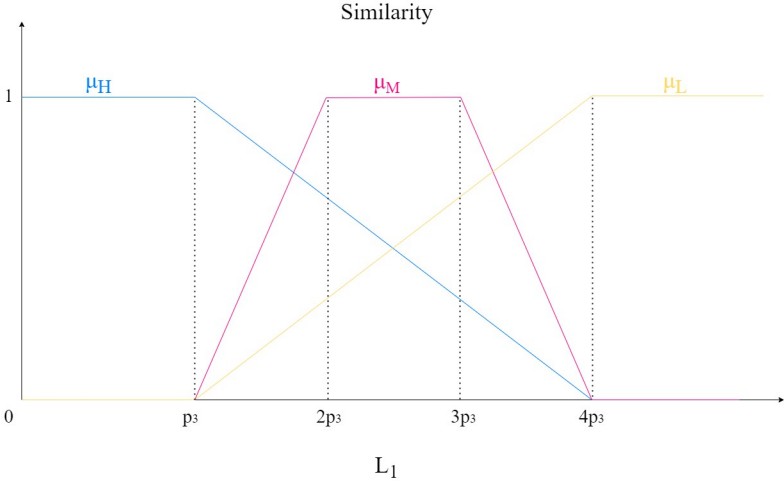

**Fig 1. Similarity degree of a pixel $x^j$ according to the distance $L_1$.**

Fuzzy Rule 2: IF ($x^j$ is not impulsive AND $x_i$ is impulsive AND $x_i$ and $x^j$ have low similarity) OR ($x^j$ is not impulsive AND $x_i$ is not impulsive AND $x_i$ and $x^j$ have high similarity) THEN $\omega_j$ is high

Fuzzy Rule 3: IF ($x^j$ is impulsive) OR IF ($x^j$ is not impulsive AND $x_i$ is impulsive AND $x_i$ and $x^j$ have high similarity) OR ($x^j$ is not impulsive AND $x_i$ is not impulsive AND $x_i$ and $x^j$ have moderate similarity) OR ($x^j$ is not impulsive AND $x_i$ is not impulsive AND $x_i$ and $x^j$ have low similarity) THEN $\omega_j$ is low

**Fuzzy coefficients.** The data output of the fuzzy system is represented with a linguistic variable. This variable represents the value of the weights in the averaging process and can take "high", "moderate" and "low" values, given by the following membership values:

$$\eta_M(w_j) = \begin{cases} \dfrac{2w_j - 1}{2*p_4 - 1} + 1, & 1 - p_4 < w_j \leq 0.5, \\[2mm] \dfrac{1 - 2w_j}{2*p_4 - 1} + 1, & 0.5 < w_j < p_4, \\[2mm] 0, & elsewhere, \end{cases} \tag{7}$$

$$\eta_H(w_j) = \begin{cases} \dfrac{w_j - 1}{1 - p_4} + 1, & p_4 < w_j \leq 1, \\[2mm] 0, & elsewhere, \end{cases} \tag{8}$$

$$\eta_L(w_j) = \begin{cases} \dfrac{w_j}{p_4 - 1} + 1, & 0 \leq w_j \leq 1 - p_4, \\[2mm] 0, & elsewhere. \end{cases} \tag{9}$$

The parameter $p_4$ was maintained as $p_4 = 0.9$ [25].

**Defuzzification.** The process of defuzzification is then carried out using the center of gravity (COG) [35] of the areas described by the trapezoids formed from the triangles given by the coefficients $\eta_H$, $\eta_M$ and $\eta_L$ and the top limit marked by the fuzzy rules, as seen in Fig 2. Let

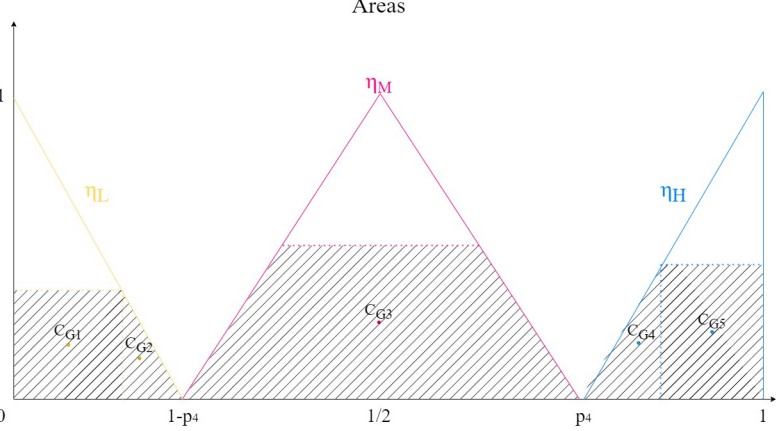

**Fig 2. Areas and COGs of a weight $\omega_j$.**

$L_j$ be the polygonal line determined by these trapezoids, then:

$$\omega_j = \frac{\int_0^1 x \cdot L_j(x)dx}{\int_0^1 L_j(x)dx}.$$

(10)

The proposed fuzzy logic method is designed to handle mixed noise, specifically impulsive noise combined with either Gaussian or Poisson noise, by applying adaptive fuzzy rules and membership functions suited to each noise type's characteristics. For Gaussian or Poisson noise (both of which typically produce a smoother, dispersed effect across the image) the fuzzy system employs similarity measures to weigh neighboring pixels. This approach averages out the background noise while preserving important structural details, ensuring that noise reduction does not blur critical edges. Impulsive noise, which appears as isolated, high-intensity pixels, is managed by calculating an impulsivity degree for each pixel; highly impulsive pixels receive lower weights in the filtering process, reducing their influence in the final image. By dynamically adjusting weights based on the unique attributes of Gaussian, Poisson, and impulsive noise, the fuzzy logic method achieves effective noise reduction in complex noise environments. This adaptability and interpretability make the method particularly valuable for medical imaging, where multiple noise types often coexist and where clarity of structural details is essential.

## Parallelization

The filtering method works pixel by pixel and is iterative, until a number of iterations is reached, which makes computational cost very high as the number of iterations increases and also as the number of images increases. Therefore, to reduce computational time, the parallel GPU version of the filter is introduced. This GPU version utilizes CUDA to parallelize the algorithm, which is a platform for computation and modeling in parallel computing developed by NVIDIA specifically for computation in graphical processing units.

The base CPU parallelization was implemented using OpenMP in [25]. It uses the *parallel for* directive in the for loop that iterates over the image pixels, allowing the processing of more pixels at once. The GPU parallelization was implemented using CUDA version 11.7 [36] and C++11. Whilst an efficient CPU parallelization could be achieved by just using a directive, GPU parallelization is not so easily achieved.

**Algorithm 1** Sequential

```
INPUT: Noisy image I, parameters n, q, r, p₁, p₂, p₃, p₄
OUTPUT: Filtered image I'
```
**1) Initialization**
```
I₀ = I
```
**for** Iteration $it = 1, \ldots$ **do**
  Image $I_{it} = I_{it-1}$
  **for** $x_i$ pixel $\in I_{it}$ **do**
    Take window $W$ $n \times n$ centered in $x_i$
    **for** $j = 1, \ldots, q$ **do**
      **2) Compute impulsivity**
      Compute $\delta(x_i)$ using Eq (3)
      **3) Compute similarity**
      Order pixels $x^j \in W$ according to $d(x_i, x^j)$
      Select the $q$ closest pixels $x^1, \ldots, x^q$
      Compute $\mu_H(x_i, x^j)$, $\mu_L(x_i, x^j)$, $\mu_M(x_i, x^j)$, using Eqs (4), (5) and (6)
      **4) Compute weights with defuzzification**

```
      Compute the fuzzy rules for {xᵢ, xʲ}
      Calculate the fuzzy coefficients for xʲ using Eqs (7), (8) and
      (9)
      Calculate weight wⱼ for xʲ using Eq (10)
    end for
    5) Calculate new xᵢ
```

$$\hat{x}_i = \frac{\sum_{j=1}^{q} \omega_j \cdot x^j}{\sum_{j=1}^{q} \omega_j}.$$

**end for**
 **end for**

In step 4 of the algorithm 1, the fuzzy rules are computed and, as seen previously, these rules require knowing the impulsivity degrees of the pixels $x^j \in W$. This means that the impulsivity degree of a pixel will often be required before it is computed in step 2. The sequential and parallel CPU implementations of the filter solve this issue by using a shared impulsivity vector where all impulsivity values are written into the first time they are needed. This solution does not work for GPU parallelization since it would need writing and reading from a kernel into a shared vector. Thus, the GPU version reorganizes the loops to separate the computation of impulsivity from the other steps. Additionally, the computation of the window $W$ is also extracted from the iterative filter and moved to initialization, since the window indexes do not vary between iterations. This makes for a less efficient sequential version due to the added *for* loops but allows for a full parallelization of the filter.

In algorithm 2, the new arrangement of the filtering process can be seen, with the sections corresponding to CUDA kernels indicated. We use three kernels in total:

- **Kernel 1:** Computes a vector with the window indexes of each pixel in the image. Each thread computes the window size of a pixel and indexes of all $x^j \in W$ in the image. Kernel 1 produces a pair of device side vectors; the first is an integer vector with all pixel indexes that compose each window; the other indicates how many pixels are in any given window. Fig 3 shows the structure of these vectors and how each thread writes into them. Fig 4 shows the code of this kernel.

- **Kernel 2:** Calculates the impulsivity degree of a pixel in the image and reorders the indexes in $W$ according to the distance of $x^j$ to $x_i$. It produces a device side vector of doubles in the range [0, 1] with the impulsivity degree of all pixels in the image in that iteration that can be passed to Kernel 3. Fig 5 shows the code of this kernel.

- **Kernel 3:** Completes the filtering process. Each thread completes steps 3, 4 and 5 in algorithm 2 for a pixel $x_i$. Fig 6 shows the code of this kernel.

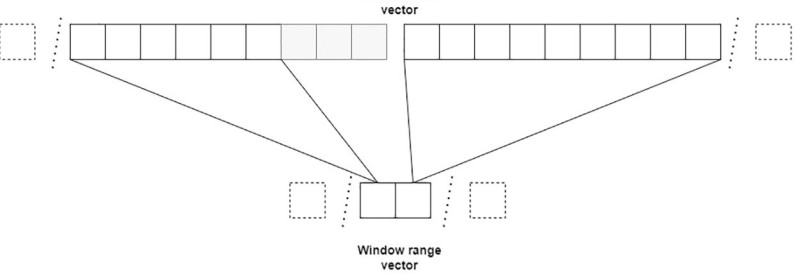

**Fig 3. Structure of the index and range vectors for a 3 × 3 window.**

```
1   __global__ void calcWindowIndexes(uint32_t* windowIndexes, uint32_t* windowRanges,
2       uint32_t rows, uint32_t cols, uint32_t windowSize, uint32_t range ){
3
4           uint32_t x =  blockIdx.x * blockDim.x + threadIdx.x;
5           uint32_t y = blockIdx.y * blockDim.y + threadIdx.y;
6           uint32_t index = cols*x+y;
7           if(x < rows && y < cols){
8               int r = 2 * windowSize + 1;
9               int c = 2 * windowSize + 1;
10
11              uint32_t lostUpperRows, lostLowerRows, lostLeftCols, lostRightCols;
12              calculateLostRows(rows, x, windowSize, lostUpperRows, lostLowerRows);
13              calculateLostCols(cols, y, windowSize, lostLeftCols, lostRightCols);
14
15              r = r - lostUpperRows - lostLowerRows;
16              c = c - lostLeftCols - lostRightCols;
17              uint32_t startingRow = x + lostUpperRows - windowSize;
18              uint32_t startingCol = y + lostLeftCols - windowSize;
19
20              int v = r*c;
21              int k = 0;
22              for(uint32_t i = startingRow; i < startingRow+range; i++)
23              for(uint32_t j = startingCol; j < startingCol+range; j++) {
24                  if(i<=startingRow+r && j<=startingCol+c){
25                      windowIndexes[index*range*range+k]=cols*i+j;
26                  }
27                  k+=1;
28              }
29              windowRanges[index] = v;
30  }}
```

**Fig 4. Kernel 1 of the GPU parallelization.**

**Algorithm 2** GPU Parallelization
INPUT: Noisy image $I$, parameters $n$, $q$, $r$, $p_1$, $p_2$, $p_3$, $p_4$
OUTPUT: Filtered image $I'$
**1) Initialization**
$I_0 = I$
**KERNEL 1**
**for** Iteration $it = 1, \ldots$ **do**
  Image $I_{it} = I_{it-1}$
  **KERNEL 2**
  **KERNEL 3**
**end for**

In Fig 7 a diagram of the iterative parallelized process is shown. To quantify the time performance of the parallelization the speed-up is utilized, defined as $S_p = \frac{T_s}{T_p}$ where $T_s$ in the computational time of the sequential method and $T_p$ the computational time of the parallel method.

```
32  __global__ void computeImpulsivity(double* imageIn, double* impulsivityDegrees,
33      uint32_t rows, uint32_t cols, uint32_t roadElements, uint32_t* windowIndexes,
34      uint32_t* windowRanges, uint32_t windowSize, double* distances ){
35
36      uint32_t x =  blockIdx.x * blockDim.x + threadIdx.x;
37      uint32_t y = blockIdx.y * blockDim.y + threadIdx.y;
38      uint32_t index = cols*x+y;
39      if(x < rows && y < cols){
40          uint maxRange = (2*windowSize+1)*(2*windowSize+1);
41          uint32_t ws = windowRanges[index];
42
43          ws = min(ws, maxRange);
44          double road_m = 0;
45
46          uint32_t elements = min(ws, roadElements);
47          road_m = calculateRoad(&windowIndexes[index*maxRange], ws, imageIn, index,
48              elements, &distances[index*maxRange]);
49
50          if(road_m <= P1) {
51              impulsivityDegrees[index] = 0;
52              //return 0;
53          } else if(road_m >= P2) {
54              impulsivityDegrees[index] = 1;
55              //return 1;
56          } else {
57              impulsivityDegrees[index] = (road_m - P1) / (P2 - P1);
58  }}}
```

**Fig 5. Kernel 2 of the GPU parallelization.**

**Memory management.** In the image version of the filter, the initialization includes computing the window indexes for each image pixel and storing them in an index vector. This is computed in a kernel outside of the iterative filtering loop. The resulting integer vector is around four times the size of the image vectors utilized, but since it was computed in the device and fits in the cache, there is not any significant impact on performance after the computation.

However, this method for window indexing is not appropriate for the volume version of the filter. Since the number of slices that compose a volume can reach the hundreds, the vector necessary to store all window indexes would no longer fit in the cache and would slow down the kernels significantly. To prevent this, the first kernel is removed, and the computation of the window indexes is moved to the other kernels, storing the window arrays in registers. As can be seen in Fig 8, this has further impact on the *fuzzyFilter* kernel. Now it has to repeat the reordering of the window by the distance, which previously was carried over from the ROAD computation in the *computeImpulsivity* kernel.

One of the challenges of GPU parallelization is the transfer of big data volumes to the GPU. To avoid the delay introduced by copying data from paged to pinned memory to prepare it for transfer to the GPU, we load the data directly in pinned memory. To ease the data load from

```
60  __global__ void fuzzyFilter(double* imageIn, double* imageOut, uint32_t* windowIndexes,
61      uint32_t* windowRanges, uint32_t windowSize, double *impulsivityDegrees,
62      uint32_t rows, uint32_t cols, uint32_t q, double* accumulator, double P3) {
63
64      uint32_t x =  blockIdx.x * blockDim.x + threadIdx.x;
65      uint32_t y = blockIdx.y * blockDim.y + threadIdx.y;
66      uint32_t index = cols*x+y;
67      if(x < rows && y < cols){
68          double oldPx = imageIn[index], newPx;
69
70          uint32_t ws = windowRanges[index];
71          uint32_t* window = windowIndexes+index*(2*windowSize+1)*(2*windowSize+1);
72
73          uint32_t localq = min(ws,q);
74          double numerator = 0;
75          double totalWeights = 0;
76
77          for(uint32_t i = 0; i < localq; i++) {
78              double pixel = imageIn[window[i]];
79              double weight = calculateWeight(index, oldPx, window[i], pixel,
80                  impulsivityDegrees, P3);
81              numerator += pixel * weight;
82              totalWeights += weight;
83          }
84
85          newPx =  numerator / totalWeights;
86
87          accumulator[index] = (originalPx - newPx) * (originalPx - newPx);
88          imageOut[index] = newPx;
89  }}
```

**Fig 6. Kernel 3 of the GPU parallelization.**

the image files to the data structures of the program, the files utilized were stored as binaries. These techniques help reduce the loading and storing time of CT volumes to under a second.

**GPU versions.** In order to take full advantage of the parallelization, a couple of variations of the third kernel were developed:

- First, a non-deterministic version of the filter denominated "chaotic" was developed. It alters the kernel, so instead of writing the new pixel values into a separate vector that is then swapped with the input vector in every iteration, the new pixels overwrite the old ones in the input image. Due to the parallelism of CUDA threads, it is not possible to determine the order the new pixels will get written in, producing different results for every execution but without greatly impacting the results.

- Another version of the filter was also developed, which skips the central pixel of the window when computing the new pixel. This version, denominated "skip-q", reduces the execution time of the threads in the kernel without significantly impacting the resulting image quality.

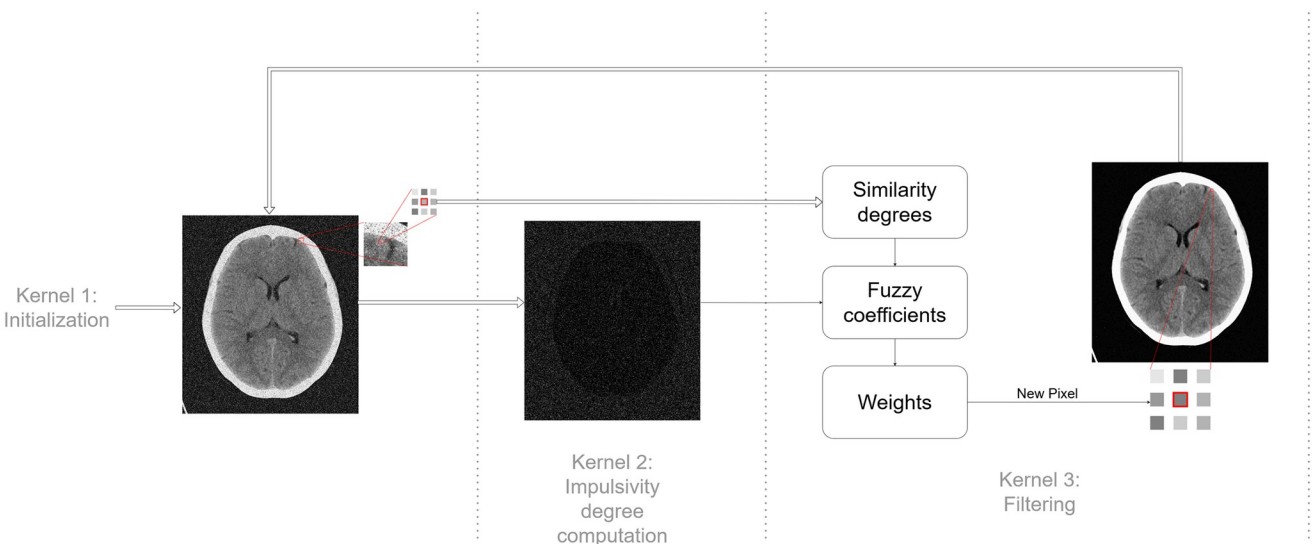

**Fig 7. Filtering process diagram.**

There was also a tested version that combined both changes, denominated "chaotic skip-q" in the experiments. The changes are trivial code-wise, but they have significant impact on the performance of the filter.

The volumetric parallelization presents a significant difference to the image version. The scale utilized is adjusted from [0, 255] proportionally to the DICOM HU re-scaled using the water attenuation coefficient. The unit change only impacts the parameters, but it does cause the filter to make significant changes in less iterations.

**Code analysis.** An analysis of the volumetric version utilizing NVIDIA Nsight compute 2022.2.1 was done on a server with an 80 GB RAM NVIDIA A100 GPU and two AMD EPYC 7282 CPUs with a total of 32 cores and 512 GB RAM.

The *computeImpulsivity* kernel was analyzed first, Fig 9 shows the roofline analysis of this kernel. The kernel is memory bound, according to the statistics given by the program the GPU Speed Of Light (SoL) Throughput is 77.30% of the maximum theoretical memory throughput. The bottleneck comes from the accesses to the L2 cache.

For the *fuzzyFilter* kernel we compared the different filter versions discussed. Fig 10 shows the roofline analysis of the kernel versions, there is an overlap between the "base" and "chaotic" versions (green borders in the graph) and the "skip-q" and combined "chaotic skip-q" versions

```
13          uint8_t ws = maxRange;
14          uint32_t windowIndexes[9];
15          double distances[9];
16          calcWindowIndexes(windowIndexes, ws, rows, cols, windowSize, (2*windowSize+1), x );
17          for(uint i = 0; i < ws; i++){
18              distances[i]=fabs(oldPx-imageIn[windowIndexes[i]+offset]);
19          }
20          bubbleSort(distances, windowIndexes, ws);
```

**Fig 8. Computation of the window in the second kernel of the volumetric implementation.**

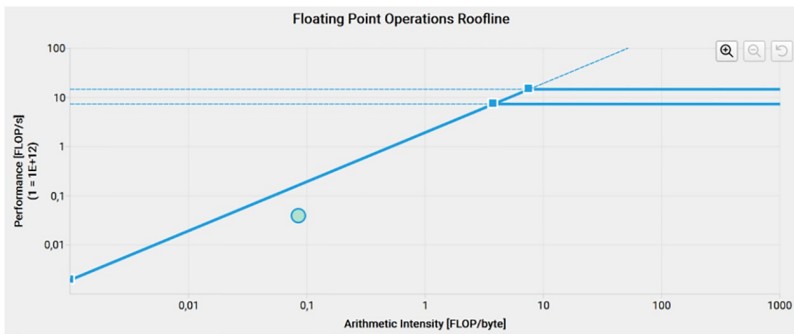

**Fig 9. Roofline of the compute impulsivity kernel.**

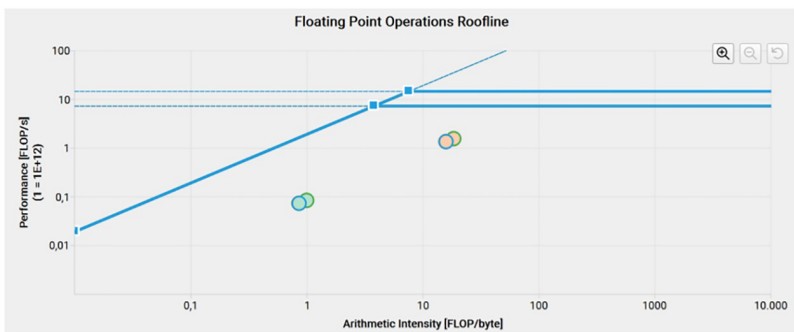

**Fig 10. Roofline of the fuzzy filter kernel.** Peach circles correspond to double precision and mint circles to single precision.

(blue borders in the graph). Whilst the single precision operations are memory bound in this kernel, the double precision operations are compute bound, although memory is still more heavily utilized. Looking at the roofline chart, the "base" and "chaotic" versions are implemented more efficiently. Fig 11 shows the GPU SoL Throughput in percentage of the maximum theoretical values. We can see here a slight decrease in throughput in the "chaotic" versions of the filter and a more significant drop in compute throughput in the "skip-q" versions, corresponding to the reduction in one iteration of the main *for* loop.

In general, for all kernels the recommendations given by Nsight Compute are the coalescence of memory accesses between the L1Tex and L2 caches (usage of 2.3 out of 4 128 bits memory blocks) and the fusing of double precision operations (estimating a theoretical increase of 30% performance increase). The window based approach of the filter causes an excessive amount of data to be loaded between caches. All elements in a row are loaded, even though only *n* elements are used. Solving this would require an overhaul of the original algorithm that is beyond the scope of this work.

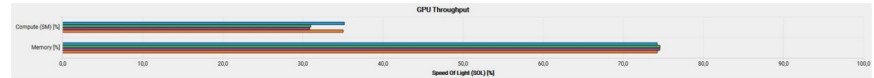

**Fig 11. SoL of the different fuzzy filter kernel versions.** The colors of the bars correspond to: blue (base version), green (skip-q version), purple (combined chaotic skip-q version) and orange (chaotic version).

## Datasets and metrics

We utilized several CT volumes, obtained from different sections of the body, and anonymized for the DICOM CT-PD Database. The volumes had been processed to obtain reconstructed images at full dose and to generate simulated low dose sinograms and reconstructed images [32]. The volumes utilized correspond to CT scans of the chest (volume C095 with 322 slices), abdomen (volume L064 with 209 slices) and head (volume N005 with 35 slices). The low dose chest scans simulate scans obtained using 10% of the original dose, while the low dose head and abdomen scans simulate 25% of the dose. Whilst both L064 and N005 are at 25% of the dose, it is necessary to take into account that the base dosage of the full dose scan is different, with Mean CTDIvol = 43.7 mGy for N005 and Mean CTDIvol = 15.6 mGy for L064. Coupled with the head scan being non-contrast and the abdomen ones being contrast-enhanced the noisiness of the low dose abdomen scan is higher than the head scan, and this reflects on the experimental results. Some of the volumes obtained at the lowest dose may not reach diagnostic quality when filtered, but they have been included in the study so as to observe the effect of the filter on very low dose and measure the quality improvement.

The header in the CT image DICOMs includes information on the water attenuation coefficient for each scan. Utilising this value, we can convert the 16-bit integers that conform the volume data to doubles which can be stored as gray-scale images in the range [0, 1], without losing information nor the capacity to re-convert the data once more to HU.

The metrics utilized to evaluate image quality are PSNR and SSIM. PSNR is an analytical image quality metric, that while easy to compute can differ from the perceived image quality. Let MAX be the maximum value an image pixel can have then the PSNR between two images $x$ and $y$ is:

$$PSNR(x, y) = 10 \log_{10}\left(\frac{MAX^2}{\left[\frac{1}{n}\sum_{i=1}^{n}(x_i - y_i)^2\right]^{1/2}}\right),$$ (11)

SSIM is a structural image quality metric, making it more adequate to measure perceived image quality. SSIM is computed as a combination of the comparisons of luminance ($l(x, y)$), contrast ($c(x, y)$) and structure ($s(x, y)$), weighted by the positive constants $\alpha$, $\beta$ and $\gamma$:

$$SSIM(x, y) = l(\mathbf{x}, \mathbf{y})^{\alpha} \cdot c(\mathbf{x}, \mathbf{y})^{\beta} \cdot s(\mathbf{x}, \mathbf{y})^{\gamma}.$$ (12)

A more extensive explanation on the computation of SSIM can be found in Brooks et al. [37].

In Table 1 we provide the mean PSNR and SSIM values for each low dose CT volume. We also provide the values for the 8-bit PGM version of the volumes since the lost information affects specially the PSNR values. When measuring the quality metrics we normalize the images in the range [0, 1] by dividing the values by the maximum value of the reference image.

**Table 1. Mean PSNR and SSIM of the unfiltered CT volumes.**

| CT Volume | PSNR | SSIM |
|---|---|---|
| C095 | 30.281 | 0.6441 |
| L064 | 47.479 | 0.9841 |
| N005 | 58.276 | 0.9983 |
| C095 (8-bit) | 30.265 | 0.6427 |
| L064 (8-bit) | 42.423 | 0.9621 |
| N005 (8-bit) | 53.697 | 0.9947 |

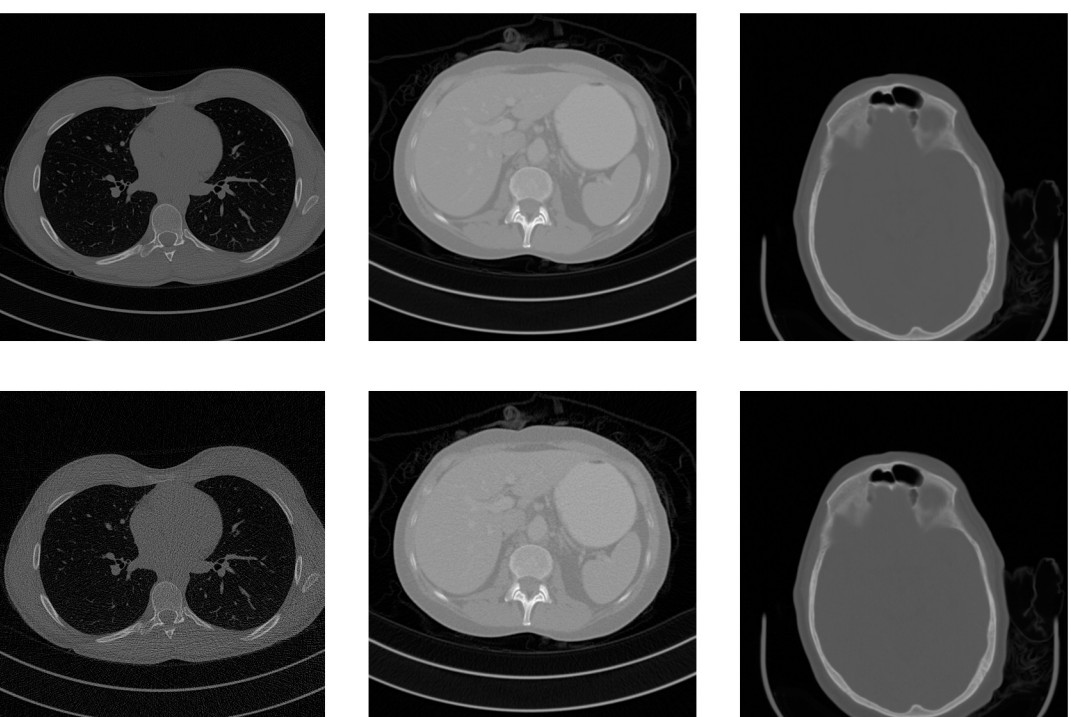

**Fig 12.** Central slice from the volumes (from right to left) C095, L064 and N005. The top row corresponds to the full dose volume and the bottom row to the lower dose.

This ensures that the images are in a range where the differences are quantifiable and that normalization does not alter the relative values of the images.

In Fig 12 we show a central slice from each of the mentioned volumes, the first row showing them at full dose and the second at the reduced dose. The image quality metrics shown in this work take as a reference the full dose volume.

Our method was implemented with CUDA version 11.7 and C++11, compiled with gcc 8.5.0. All experiments were carried out on the machine mentioned in Code analysis.

## Results

In this section, we utilized CT volumes from the DICOM CT-PD database from the Mayo clinic to evaluate the performance of the proposed methods. At first, we evaluated the different versions of the parallel filter, to determine the best in terms of quality and velocity. Next, we compared the performance of that version against other state-of-the-art filters, on the same database. Lastly, we realized a qualitative assessment of the CT volumes filtered with different stop criteria, to verify the validity of the filter for dose reduction.

### Computational efficiency analysis

We used two chest CT scans to study the different versions of the GPU parallel filter in terms of speed. A sequential version of the filter was also evaluated, so that we could determine the speed-up caused by the parallelization.

The filters were all executed with five fixed iterations, each one was executed ten times, and the filtering time was measured. The measures considered for the speed-up correspond just to

**Table 2. Execution time and speed-up of the different filter versions.**

| | | | Time-to-result (s) | Execution Time (s) | Time/Iter. | Speedup Seq. Kernels | CPU |
|---|---|---|---|---|---|---|---|
| C095 | GPU | Sequential | 942.10 | 941.78 | 188.36 | | |
| | | Base | 1.12 | 3.47E-01 | 6.94E-02 | 2712.26 | 607.18 |
| | | Chaotic | 1.12 | 3.43E-01 | 6.85E-02 | 2747.78 | 615.13 |
| | | Skip-q | 1.12 | 3.46E-01 | 6.92E-02 | 2721.12 | 609.16 |
| | | Chaotic skip-q | 1.12 | 3.41E-01 | 6.83E-02 | 2758.22 | 617.47 |
| | CPU | 1 thread | | 210.83 | 42.17 | | |
| | | 32 threads | | 24.82 | 4.96 | | 8.49 |
| C012 | GPU | Sequential | 1042.66 | 1042.19 | 208.44 | | |
| | | Base | 1.21 | 3.79E-01 | 7.58E-02 | 2750.59 | 556.44 |
| | | Chaotic | 1.21 | 3.74E-01 | 7.48E-02 | 2786.90 | 563.78 |
| | | Skip-q | 1.21 | 3.78E-01 | 7.55E-02 | 2759.58 | 558.26 |
| | | Chaotic skip-q | 1.21 | 3.73E-01 | 7.45E-02 | 2797.20 | 565.87 |
| | CPU | 1 thread | | 226.52 | 45.30 | | |
| | | 32 threads | | 27.27 | 5.45 | | 8.31 |

the iterative filtering loop, the time for pre- and post-processing was also measured, but separately.

The CPU parallelization was executed differently, since the only existing version works on 8-bit images and not on volumes. The filter was executed for all slices with five fixed iterations, and afterwards the time measured for the iterative filtering loop of each image was added together to get the volume filtering time. The process was carried out with one and thirty-two threads, so as to measure the speedup the parallelization achieves over the sequential version of the same program.

In Table 2 we show these execution times for the sequential and parallel versions of the filter as well as the speed-up calculated for each parallel version as defined in section Materials and methods. Additionally, the time-to-result column has been added to the GPU methods, which includes load and store times. All GPU versions achieve considerable speedup compared to their sequential versions, with the combination of the non-deterministic and "skip-q" versions achieving the highest speed-up. On the other hand, the CPU parallelization is far from the ideal speedup expected for thirty-two threads, this is caused due to the overhead introduced when creating the threads being significant when the time it takes to filter an 8-bit slice of 512×512 pixels is well under a second.

We have also obtained measures for average PSNR and SSIM of the C095 volume filtered by the GPU parallel methods, shown in Table 3. These do not correspond with an optimized

**Table 3. Average PSNR and SSIM resulting from the parallel GPU versions executed with five iterations on volume C095.**

| | PSNR (dB) | SSIM |
|---|---|---|
| Unfiltered | 30.281 | 0.644 |
| Base GPU | 35.062 | 0.856 |
| Chaotic GPU | 34.972 | 0.856 |
| Skip-q GPU | 34.747 | 0.853 |
| Chaotic skip-q GPU | 34.623 | 0.853 |
| Base CPU | 33.233 | 0.767 |

**Table 4. PSNR and SSIM in a slice of C095 and of the Phantom filtered with the bilateral filter and different neighborhood sizes.**

| NeighborhoodSize | C095 | | Phantoma (Gaussian noise) | |
|---|---|---|---|---|
| | PSNR (dB) | SSIM | PSNR (dB) | SSIM |
| 3 | 23.223 | 0.6341 | 28.870 | 0.3033 |
| 5 | 23.279 | 0.6441 | 29.451 | 0.3230 |
| 7 | 23.284 | 0.6450 | 29.494 | 0.3243 |
| 9 | 23.284 | 0.6451 | 29.495 | 0.3244 |
| 11 | 23.284 | 0.6451 | 29.495 | 0.3244 |
| 13 | 23.284 | 0.6451 | 29.495 | 0.3244 |
| 15 | 23.284 | 0.6451 | 29.495 | 0.3244 |

usage of the filter, but they indicate the difference in performance between the filters. In the following experiments, we use exclusively the non-deterministic versions of the filters, as quality wise they do not differ majorly from the deterministic versions and they are faster.

## Comparative quality analysis with other arithmetic filters

The two arithmetic methods we are comparing the FGI filter with have shown very good performance with medical images. The FPGA is also based on fuzzy logic and presented good performance with simulated mixed Gaussian-impulsive noise. The arguments used in the filter are the same as in [28]. The bilateral filter used is the version integrated in Matlab. We've set the argument "NeighborhoodSize" to 9, to determine this value we tested the filter on a slice of volume C095 and of the Phantom with values from 3 to 15, as seen in Table 4. We are comparing these two filters with the chaotic version of the GPU parallel filter. Since we are not using any quality metrics as stop criteria we executed the FGI method twenty times, with one to twenty iterations, as we can not know a priory how many iterations are ideal for each noise type and volume.

In Table 5 we show the average PSNR and SSIM obtained by each method for the Shepp-Logan phantom [38] with added noise, excluding the slices that contain only air. The filters were evaluated with Gaussian noise at two different variances, Gaussian-impulsive noise, poisson noise and impulsive noise. For the FGI method, we have shown the values corresponding to the best SSIM result for each noise type, resulting in fifteen iterations for all noise types except poisson, which was best at eleven iterations. The FGI method shows better performance than the other two methods in most cases, achieving the highest SSIM values for all noise

**Table 5. Average PSNR and SSIM of the different filtering methods on the Shepp-Logan phantom with added noise.**

| | | No filter | Bilateral | FPGA | FGI method |
|---|---|---|---|---|---|
| Gaussian noise ($\sigma^2$=0.0005) | PSNR | 34.980 | 42.093 | 31.090 | 30.110 |
| | SSIM | 0.5757 | 0.7769 | 0.8240 | 0.8570 |
| Gaussian noise ($\sigma^2$=0.005) | PSNR | 24.996 | 29.807 | 28.056 | 28.052 |
| | SSIM | 0.1551 | 0.2846 | 0.3421 | 0.4400 |
| Gaussian-impulsive noise ($\sigma^2$=0.0005) | PSNR | 16.373 | 16.463 | 25.865 | 28.671 |
| | SSIM | 0.1780 | 0.2190 | 0.7205 | 0.7867 |
| Poisson noise | PSNR | 25.760 | 32.331 | 24.476 | 24.450 |
| | SSIM | 0.9120 | 0.9854 | 0.9664 | 0.9742 |
| Impulsive noise | PSNR | 16.437 | 16.478 | 26.001 | 29.184 |
| | SSIM | 0.2626 | 0.2709 | 0.8564 | 0.9217 |

types. In Fig 13 we show slice 165 from the phantom with the different added noise types before and after filtering with the three methods.

To evaluate the quality improvement in a real case we also filtered volumes C095, L064 and N005, in Fig 14 we show a central slice from the C095 volume, before and after filtering with each of the methods. We chose ROIs, shown in Fig 15, in a number of slices of each of the volumes and computed the standar deviation (SD) in them, the graphs in Fig 16 show these SD values for the full dose images, unfiltered low dose images and the low dose images filtered with the bilateral filter, FPGA and the new FGI method, Table 6 shows their average SD value. All methods achieve lower SD values than the reference full dose images for volume L064 and FPGA and the FGI method do as well for volume N005, but for volume C095 only FGI achieves lower SD values than the reference.

## Comparative quality analysis with AI-based filters

Since the pre-trained models were trained with the volumes from the 2016 low dose CT Grand Challenge [39], which consisted of contrast enhanced abdominal scans, we carried out the experiments with volumes from the DICOM-CT-PD database that were also contrast enhanced abdominal scans. We used volume L064, reconstructed with FBP at a quarter (QD) and an eighth (ED) of its regular dose, and compared PSNR and SSIM metrics. The ED volume was achieved by reconstructing the QD scan with half of the projections.

Metrics were taken in a central ROI of size $256 \times 256$, as can be seen in Fig 17 on a central slice of the QD and ED volumes.

Since the images from the CNCL method are in a different format, we employed two different normalization methods. First a zero-means normalization ("Zero-means" in the tables), which is a normalization over the image itself. The second method is a normalization of values by adjusting them to the range of the reference image by dividing by its MAX value ("/Max" in the tables), this method keeps the relative gray values between images when adjusting the range. We computed the metrics on the unfiltered PNG and attenuation coefficient volumes, the results can be consulted in Table 7.

The resulting metrics for the volumes filtered with WavResNet and FGI can be consulted in Table 8, with the "/Max" normalization. The metrics for the CNCL filtered images and the FGI volume, with the "Zero-means" normalization, can be consulted in Table 9. The FGI filter was executed for a single iteration for the QD volume and three iterations for the ED volume, we utilized the chaotic version of the filter in this experiment. Although very close, FGI achieves better results for PSNR than WavResNet and both are equals in SSIM. It also achieves better results than CNCL for both metrics, even when we compare the improvement over their respective unfiltered volumes instead. The filter results can be seen on a central slice of QD and ED L064 in Fig 18.

Through the use of membership functions and rule-based decision-making, the proposed fuzzy logic system can adjust seamlessly to varying noise levels, offering a customized approach for each noise type. This flexibility contrasts with machine learning methods, which typically require large datasets and extensive training to manage noise effectively. While machine learning models, particularly deep learning algorithms, can perform well under diverse noise conditions, they often function as opaque black boxes, limiting their interpretability. In contrast, the rule-based nature of fuzzy logic makes each filtering step traceable and clear, an essential quality in medical imaging where transparency and reliability are paramount. Additionally, fuzzy logic tends to handle outliers and atypical noise patterns robustly without the need for retraining, unlike machine learning models, which may require frequent recalibration to retain accuracy across different noise scenarios.

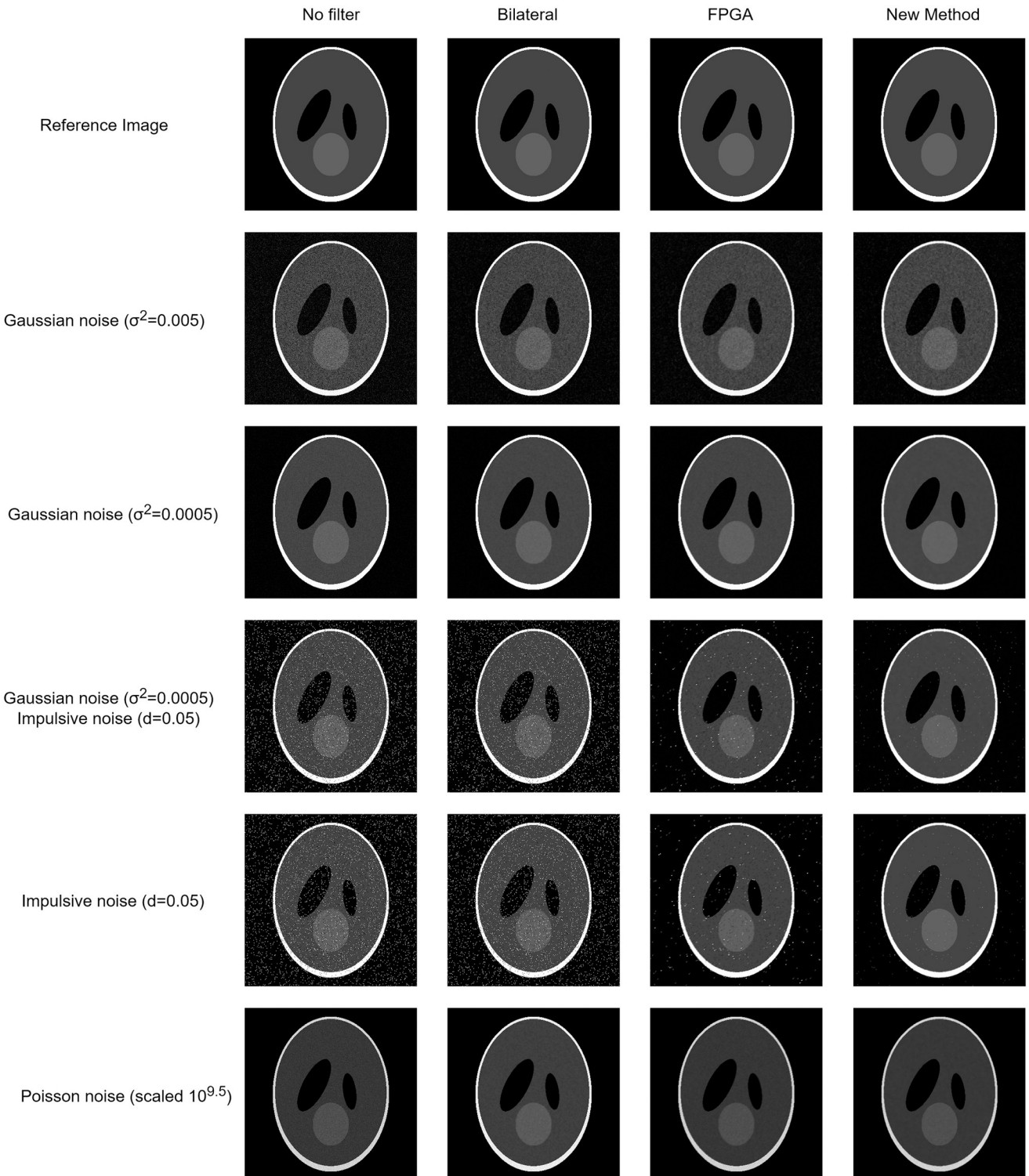

**Fig 13. Slice 165 from the Shepp-Logan phantom with different added noise.** From left to right: unfiltered, filtered with the bilateral filter, the FPGA and the FGI method.

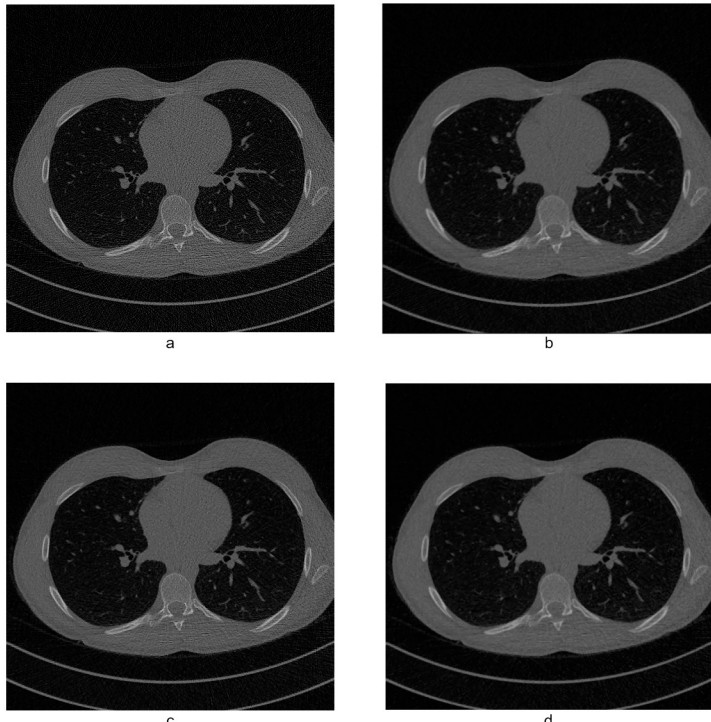

**Fig 14. Central slice from the volume C095.** a. Unfiltered, b. FPGA, c. Bilateral, d. FGI method.

## Qualitative study

We conducted a qualitative assessment on the efficacy of the FGI filter, regarding several criteria: image noise (texture of anatomical structures), sharpness (representation of anatomic details), image quality (presence of artifacts, contrast, and spatial resolution). All features were evaluated using a 5-point Likert scale: 5 = excellent, 4 = good, 3 = moderate, 2 = acceptable, and 1 = suboptimal. Window width and window level were changed as would be done during routine image interpretation.

To carry out this evaluation, we obtained CT volumes filtered with the "chaotic" and "chaotic skip-q" versions of the FGI filter utilizing the quality metrics, maximum PSNR or

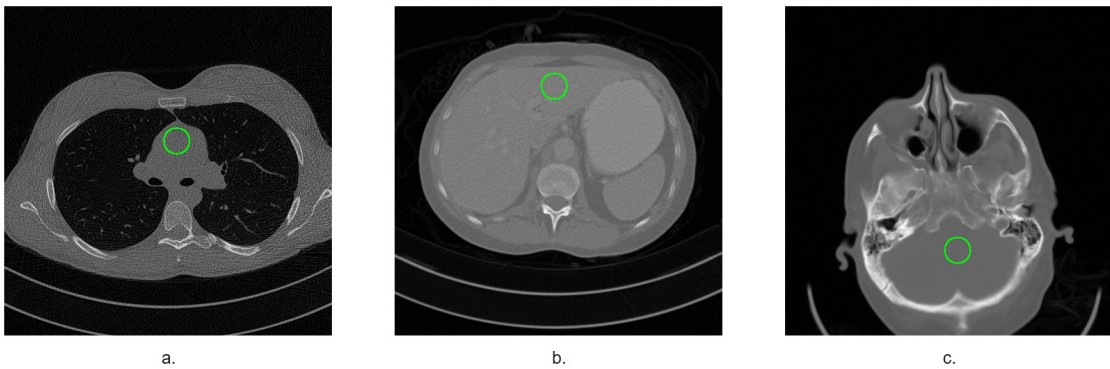

**Fig 15. A slice from each volume showing the ROI chosen.** a. C095, b. L064, c. N005.

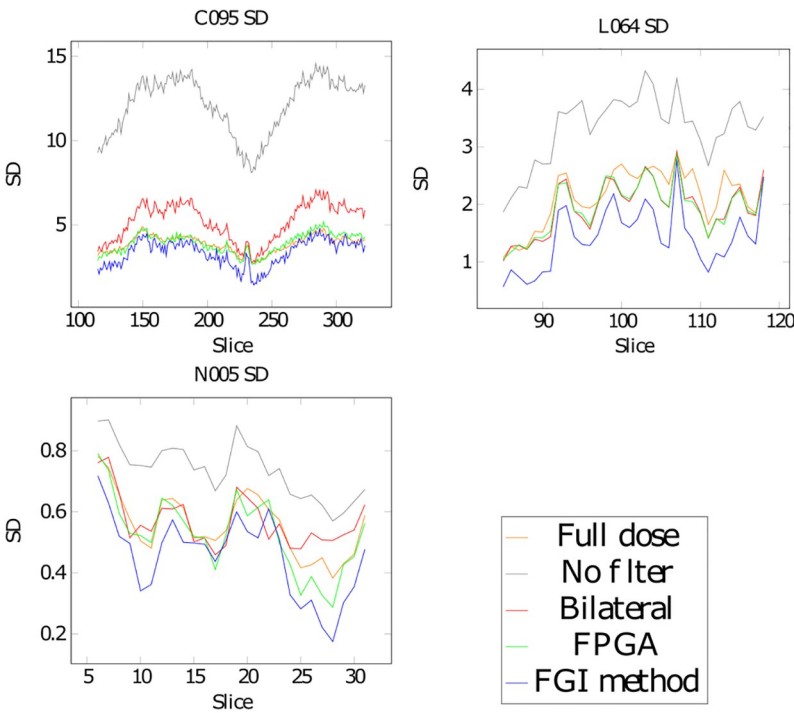

**Fig 16. SD for a set of slices of volumes C095, L064 and N005.**

**Table 6. Average SD of the different filtering methods on volumes C095, L064 and N005.**

| | Full dose | No filter | Bilateral | FPGA | FGI method |
|---|---|---|---|---|---|
| C095 | 3.888 | 12.117 | 5.200 | 3.919 | 3.343 |
| L064 | 2.236 | 3.392 | 2.030 | 2.032 | 1.520 |
| N005 | 0.558 | 0.737 | 0.570 | 0.526 | 0.454 |

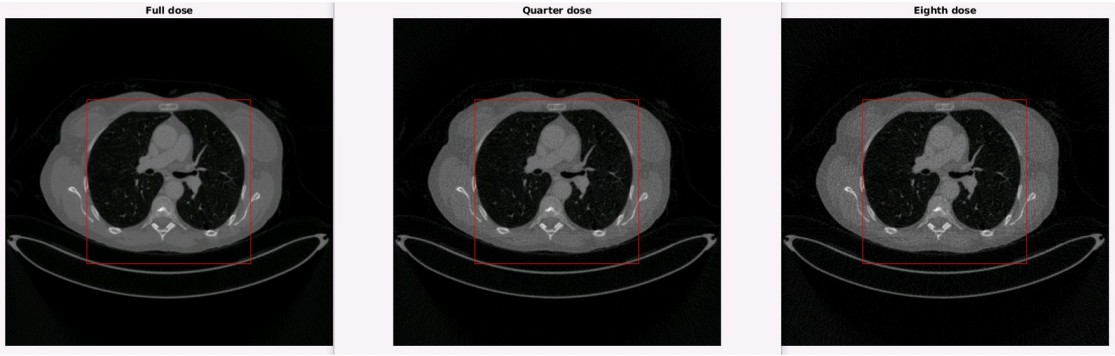

**Fig 17. ROIs in a central slice of full dose (FD), QD and ED L064.**

**Table 7. PSNR and SSIM of the QD and ED unfiltered volume L064.**

| | PNG | | Attenuation | | | |
|---|---|---|---|---|---|---|
| | PSNR Normalized | SSIM Normalized | PSNR /Max | SSIM /Max | PSNR Normalized | SSIM Normalized |
| QD | 33.959 | 0.9996 | 40.166 | 1.0000 | 34.874 | 0.9996 |
| ED | 30.928 | 0.9991 | 36.876 | 1.0000 | 32.114 | 0.9991 |

**Table 8. PSNR and SSIM of the QD and ED L064 filtered with WavResNet and FGI, normalized with /Max.**

| | WavResNet | | FGI | |
|---|---|---|---|---|
| | QD | ED | QD | ED |
| PSNR /Max (dB) | 41.315 | 37.947 | 41.764 | 39.343 |
| SSIM /Max | 1.0000 | 1.0000 | 1.0000 | 1.0000 |

**Table 9. PSNR and SSIM of the QD and ED L064 filtered with CNCL and FGI, with zero-means normalization.**

| | CNCL | | FGI | |
|---|---|---|---|---|
| | QD | ED | QD | ED |
| PSNR Zero-means (dB) | 33.304 | 30.836 | 36.076 | 33.618 |
| SSIM Zero-means | 0.9994 | 0.9989 | 0.9997 | 0.9994 |

maximum SSIM, as stopping criteria (SC). In total we evaluated four different filtered versions of each of the three volumes, in Table 10 we show the average PSNR and SSIM of each of these versions. The quality metrics indicate very high quality for all filtered versions of the three volumes, with the stop criteria of reaching maximum PSNR giving better results in all cases. The chest volume, which had the highest amount of noise, has a SSIM value under 0.9 in all cases which indicate that there is a perceptible difference between the full dose volume and the filtered volumes. On the other hand, the head volume has the same average quality as the low dose volume for all filtered versions, so any differences could only be detected by an observer. A clinician evaluated all volumes, their findings can be found in Table 11.

## Discussion

On the roofline analysis carried out with Nsight we observed that the FGI filter is mostly memory bound, with only the double precision operations in the *fuzzyFilter* kernel being compute bound. This indicates that a more efficient memory management and a reduction of memory accesses are needed for a more efficient filter.

This is corroborated by the results obtained on the efficiency analysis. The volumetric GPU parallel filter achieves considerable speedup compared to the sequential versions, with the combination of the non-deterministic (less readwrite operations) and "skip-q" (less memory accesses) versions achieving the highest speedup at 2797 for volume C012 when comparing it to its sequential version and 617 for volume C095 when comparing it to the CPU version.

Quality-wise, the non-deterministic versions do not differ significantly from the deterministic versions, but the skip-q versions do show worse results. The quantitative study showed higher PSNR and SSIM values for the "base" and "chaotic" versions. The qualitative study conducted also reached similar results, when comparing "chaotic" and "chaotic skip-q" versions, the "chaotic" consistently achieved the higher scores amongst the filtered volumes.

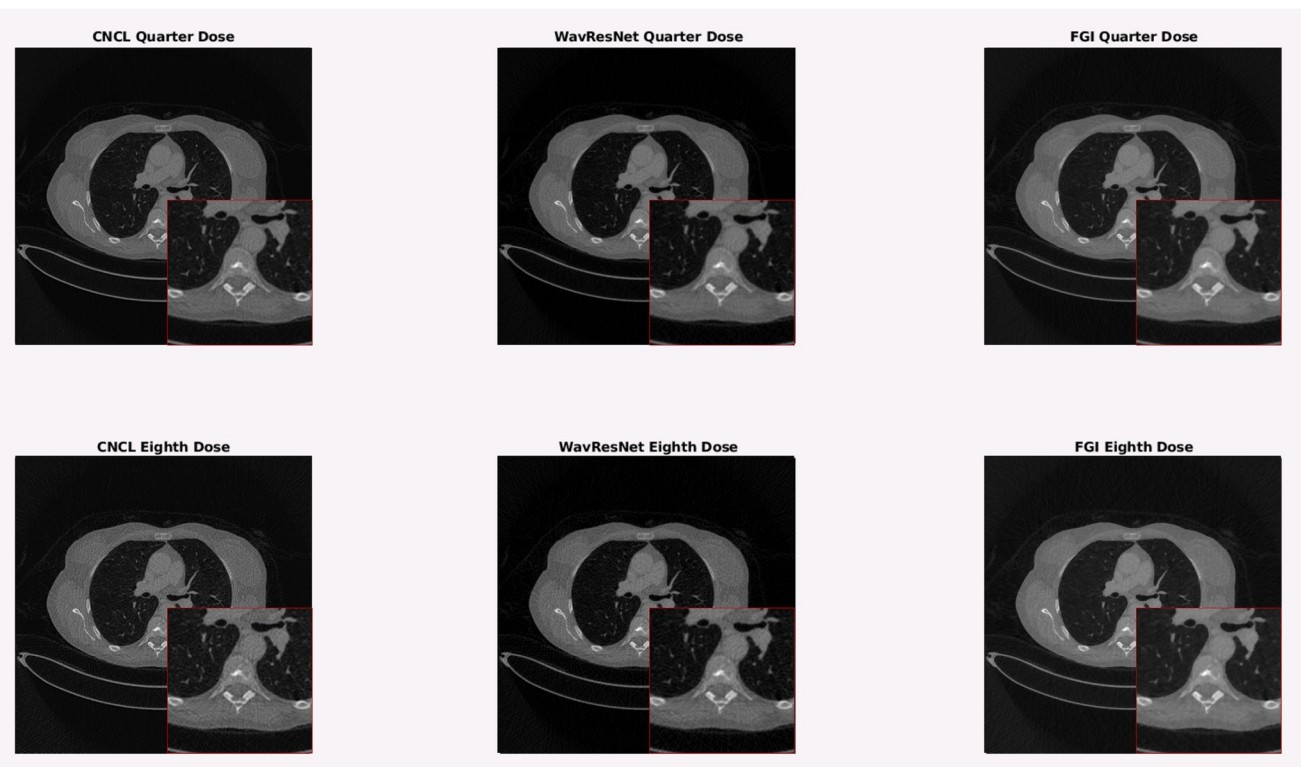

**Fig 18. QD and ED L064 filtered with CNCL, WavResNet and FGI.**

**Table 10. Average PSNR and SSIM of the volumes with different stop conditions (SC).**

|  |  | C095 | | L064 | | N005 | |
|---|---|---|---|---|---|---|---|
|  |  | PSNR | SSIM | PSNR | SSIM | PSNR | SSIM |
| Chaotic | SC max PSNR | 35.7676 | 0.8632 | 47.3248 | 0.9846 | 58.2757 | 0.9983 |
|  | SC max SSIM | 35.7555 | 0.8646 | 47.2892 | 0.9846 | 58.2757 | 0.9983 |
| Chaotic skip-q | SC max PSNR | 35.6788 | 0.8620 | 47.4793 | 0.9841 | 58.2757 | 0.9983 |
|  | SC max SSIM | 35.6689 | 0.8638 | 47.4788 | 0.9842 | 58.2757 | 0.9983 |

**Table 11. Clinician evaluation of the different volumes in a 5-point Likert scale.**

|  | C095 | | | L064 | | | N005 | | |
|---|---|---|---|---|---|---|---|---|---|
|  | NOISE | SHARPNESS | QUALITY | NOISE | SHARPNESS | QUALITY | NOISE | SHARPNESS | QUALITY |
| FULL | 3 | 3 | 3 | 5 | 5 | 5 | 5 | 5 | 5 |
| LOW | 1 | 1 | 2 | 4 | 5 | 4 | 4 | 4 | 4 |
| PSNR_CH | 1 | 2 | 2 | 4 | 5 | 4 | 4 | 4 | 4 |
| PSNR_CHSQ | 2 | 2 | 2 | 3 | 5 | 4 | 3 | 4 | 4 |
| SSIM_CH | 2 | 2 | 2 | 4 | 5 | 4 | 3 | 4 | 4 |
| SSIM_CHSQ | 2 | 2 | 2 | 3 | 5 | 4 | 3 | 4 | 4 |

The tests on the simulated noise in the Shepp-Logan phantom showed that the FGI filter is capable of restoring the structural integrity of the image, achieving high SSIM values for most noise types. However, the filter is not designed to restore gray values outside of unifying the different values in a structure, as demonstrated by the lower PSNR values for the Gaussian and Poisson noise types.

The Nsight analysis and the performance achieved in the results are valid for the characteristics of the machine utilized. There exist limitations when using the method in machines with different capabilities, especially with lower GPU memory, as the method is mostly memory bound. It could pose a problem in real-scenario applications, since hospital machines are not expected to have state of the art GPUs. However, these limitations affect computational speed but do not compromise image quality.

A limitation of the method is that previous to its implementation in a specific machine it would be necessary to carry out a heuristic study to determine the optimal number of iterations to maximize image quality. A variation in the number of iterations would also affect computational speed. However, a slight variation over the optimal would be in the range of milliseconds for computational speed and would not affect image quality significantly.

Another aspect to consider when applying the method is precision. When testing a different precision, single instead of double, it was found that it did not have a noticeable impact in image quality and the impact in performance was not significant. Since the dataset DICOM data is originally stored in unsigned 16-bit integers, both single and double precision floating-point numbers are precise enough to represent the corresponding attenuation coefficients. Nevertheless, the kernels in single precision are still memory bound, so impact in performance is minimal. Additionally, double precision operations were used in the tests, if the filter was involved in the reconstruction process double precision would be expected, as precision has significant impact on reconstruction quality [40].

GPU-based acceleration techniques can be found in nearly all imaging modalities, as they provide faster computations than CPU, leading to real-time implementations [41], several techniques can be used to further optimize GPU-based applications. Pinned memory was used in this work to accelerate the load of data into GPU. Nikitin [42] used this technique as well in the GPU-based tomographic reconstruction library in order to run data transfers concurrently to GPU computations. Lastly, Quintana et al. [43] used it to accelerate data transfers between main memory and the GPU, as well as between disk and main memory.

Further optimization can be achieved with the use of multiple GPUs. Heterogeneous GPUs were used by Chou et al. [44] to resolve the memory-bound issue that appeared with iterative reconstruction at ultrahigh resolution in a single GPU. Similarly, Wu et al. [45] utilize multi-GPUs to allow for the reconstruction of high-resolution CT volumes in real-time. Additionally, in that work the authors use several key features of modern GPUs, GPUDirect RDMA, GPUDirect Storage and Tensor Cores, which were used to further optimize backprojection when it is compute-bound.

Very low-dose scans present an amount of noise and artifacts that prove difficult to eliminate with the proposed filter. The recommendations for such high-noise is to eliminate noise before reconstruction, but when this is not possible there are other techniques that can be applied. Niwa et al. [46] proposes an image-based forward projection and filtering of the artificial sinogram to reduce streak artifacts, while the work uses a Gaussian filter the FGI filter could be used instead. A different approach, that we will explore in future work, is the use of 3D windows with the idea of computing the areas obscured by artifacts by referencing the surrounding slices.

## Conclusion

In this paper, we introduced several parallel methods based on fuzzy logic for filtering CT medical images. These methods were implemented on GPUs using CUDA. We studied their computational performance compared to the sequential versions and how they compare to the performance achieved with CPU parallelization. The image quality was evaluated quantitatively amongst the methods and against the FPGA and bilateral filters. A qualitative evaluation of the filter was also carried out by a radiologist.

An analysis of the volumetric parallelization with Nsight Compute showed that the "base" and "chaotic" methods use the computing capabilities of the testing environment more efficiently. The analysis also revealed that the algorithm is limited in its memory access efficiency due to the window based approach. In the future we will study a block based approach to the algorithm to solve this issue.

Speed-wise, the GPU parallelization is better than the CPU parallelization in all cases. It is significantly better suited for volumes. All the different versions achieve good performance, but the non-deterministic versions are faster. The reduction of the elements to iterate through in the "skip-q" versions also reduces execution time, thus the fastest GPU version is the non-deterministic "chaotic skip-q".

Volume filtering time is under 0.1 seconds per iteration for all GPU versions, so we can consider that real-time filtering is achieved.

For simulated low-dose CT images, good image quality can be achieved in under five iterations, with the "base" and "chaotic" versions achieving the best results. When filtering the simulated noise in the Shepp-Logan phantom, the FGI filter shows better performance than the FPGA and bilateral filters in most cases, achieving the highest SSIM values for most noise types. When comparing SD values on real CT volumes, the FGI is the best for all volumes. For 10% dose it was the only one capable of achieving SD values close to the full dose.

The filter demonstrates competitive results when compared to the AI based filters CNCL and WavResNet, achieving higher PSNR and SSIM values than both of them when tested on contrast enhanced abdominal volumes at 25% and 12.5% of dose.

The qualitative study, evaluated by a clinician, showed that the filter is capable of increasing the quality of the filtered 10% dose volume from "suboptimal" to "acceptable". Right now, it is a post-processing filter, but it would be interesting to study its integration in iterative reconstruction methods, as it could lead to more significant results.

For real environment integration, it would be interesting to evaluate the efficiency of the filter on machines with the characteristics of the hospital computers that would perform these operations. We can predict the need for memory optimization since the implementations currently rely on large cache and register sizes.

## Author Contributions

**Conceptualization:** Celia Tendero Delicado, Josep Arnal García, Vicent Vidal Gimeno.

**Data curation:** Mónica Chillarón Pérez.

**Formal analysis:** Josep Arnal García.

**Funding acquisition:** Vicent Vidal Gimeno.

**Investigation:** Celia Tendero Delicado, Mónica Chillarón Pérez.

**Methodology:** Celia Tendero Delicado, Josep Arnal García, Vicent Vidal Gimeno, Esther Blanco Pérez.

**Project administration:** Vicent Vidal Gimeno.

**Resources:** Mónica Chillarón Pérez.

**Software:** Celia Tendero Delicado.

**Supervision:** Josep Arnal García, Vicent Vidal Gimeno.

**Validation:** Mónica Chillarón Pérez, Josep Arnal García, Esther Blanco Pérez.

**Writing – original draft:** Celia Tendero Delicado.

**Writing – review & editing:** Mónica Chillarón Pérez, Josep Arnal García, Vicent Vidal Gimeno.

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
