## [Decision Letter · Decision Letter 0]

29 Oct 2024

PONE-D-24-40384A Gpu-Accelerated Fuzzy Method for Real-Time CT Volume FilteringPLOS ONE

Dear Dr. Tendero Delicado,

Thank you for submitting your manuscript to PLOS ONE. After careful consideration, we feel that it has merit but does not fully meet PLOS ONE’s publication criteria as it currently stands. Therefore, we invite you to submit a revised version of the manuscript that addresses the points raised during the review process.

We look forward to receiving your revised manuscript.

Kind regards,

Khan Bahadar Khan, Ph.D

Academic Editor

PLOS ONE

Journal Requirements:

“This research has been supported by the TED project Grant Reference TED2021-131091B-I00 funded by MCIN/AEI/ 10.13039/501100011033 and by the “European Union NextGenerationEU/PRTR”. Funding for open access charge: CRUE-Universitat Politècnica de València.”

“This research has been supported by the TED project Grant Reference TED2021-131091B-I00 funded by MCIN/AEI/ 10.13039/501100011033 and by the “European Union NextGenerationEU/PRTR”.”

“This research has been supported by the TED project Grant Reference TED2021-131091B-I00 funded by MCIN/AEI/ 10.13039/501100011033 and by the “European Union NextGenerationEU/PRTR”. Funding for open access charge: CRUE-Universitat Politècnica de València.”

Reviewers' comments:

Reviewer's Responses to Questions

**Comments to the Author**

1. Is the manuscript technically sound, and do the data support the conclusions?

Reviewer #1: Yes

Reviewer #2: Yes

2. Has the statistical analysis been performed appropriately and rigorously? 

Reviewer #1: Yes

Reviewer #2: Yes

3. Have the authors made all data underlying the findings in their manuscript fully available?

Reviewer #1: Yes

Reviewer #2: No

4. Is the manuscript presented in an intelligible fashion and written in standard English?

Reviewer #1: No

Reviewer #2: Yes

5. Review Comments to the Author

Reviewer #1: Dear Authors

The manuscript reflects good research effort. However, it deserves more time in arranging the text (Introduction, Material, Method (including Metrics), Discussion and conclusion. This would ease understanding and following the finding effectively. for examples, please consider the followings:

I- Moving the Material and metrics subsection to the Materials and method section (in the current text we have two same headings : 1-) Materials and Method 2-) Material and Metrics. why?

II_ Adding a Table describing the number of CT volumes you experimented for each dataset (C095, L064, N005), or you mention them in the text clearly in the Material section.

III- Are there limitations to discuss with audience (or there are no limitations at all !!??) . Short paragraph in the discussion would be adequate.

IV- How you would expect the performance of the method if variable are changed (e.g. number of iterations, or computer hardware capabilities). Couple of sentences/statements in the discussion section would be adequate.

V- Where is the table you mentioned in the last sentence in page 19 qualitative study subsection?

VI- Finally, the reference list need some update work to be mentioned and discussed (how your method is different from them). Here are some examples that may be considerable of all of them or only the GPU ones at least :

https://journals.iucr.org/s/issues/2023/01/00/mo5264/mo5264.pdf

https://www.mdpi.com/1424-8220/24/6/1947

https://www.sciencedirect.com/science/article/pii/S0169260722001110

https://www.nature.com/articles/s41598-023-46028-9

https://www.sciencedirect.com/science/article/pii/S0010465523000644

https://dl.acm.org/doi/abs/10.1145/3650200.3656634

https://www.osti.gov/biblio/2429797

https://www.mdpi.com/2079-9292/10/24/3118

The above links are from Google Scholar published papers since 2020, you may search to find more in the topic. A paragraph or two in the discussion section would be adequate (i.e. comparing your method with the literature).

In summary, the discussion section in the manuscript needs extensions based on the concerns above (III, IV, and VI).

Reviewer #2: This paper proposes an innovative method for filtering medical CT volumes in real-time using GPU parallelization and fuzzy logic. The method addresses the issue of noise in CT images, particularly mixed Gaussian-impulsive noise, which often degrades image quality. The authors develop a new fuzzy filter capable of denoising CT images with high speed and accuracy by utilizing parallel computing on GPU architectures, specifically NVIDIA’s CUDA platform. The paper presents significant speedups (over 2700x) and compares the quality of the proposed filter with existing state-of-the-art techniques like Bilateral Filter, WavResNet, and CNCL. The experiments demonstrate that the proposed method achieves competitive performance in terms of PSNR, SSIM, and processing time. This approach offers a significant contribution by enabling real-time denoising of large CT volumes, which is critical for reducing radiation exposure in medical imaging.

This paper might be of interest to the research community focused on medical imaging, particularly in the domains of image denoising and parallel computing. The need for real-time denoising solutions in medical CT imaging is critical due to the implications for reducing patient exposure to radiation while maintaining diagnostic quality. The combination of fuzzy logic with GPU acceleration is a novel approach that could inspire further research on parallelization techniques in medical imaging.

Although the topic is of interest and results are promising, there are some issues that should be addressed prior publication:

1. Introduction section should include a more explicit hypothesis, such as: “This paper hypothesizes that a GPU-accelerated fuzzy filter can achieve both real-time processing and superior noise reduction in low-dose CT images compared to existing methods.”

2. Please expand the literature review to include more recent developments in deep learning for medical image denoising, such as the use of convolutional neural networks (CNNs) and adversarial networks in CT image processing.

3. *While the technical description is thorough, the explanation of certain parameters (e.g., the fuzzy membership functions and the specific settings for CUDA optimization) could be more detailed. This is important for other researchers attempting to replicate the results.

4. My main concern about this research work is that the novelty is mainly technological, focusing on speedup and parallelization. A deeper discussion of how fuzzy logic enhances noise filtering in medical images, beyond computational efficiency, would strengthen the case for novelty.

5. While the method is evaluated on multiple datasets, the results could be strengthened by testing with additional, more diverse types of CT scans or by including additional performance metrics (e.g., time-to-result vs. hardware resource usage).

6. Discuss in more detail the scenarios where the filter does not perform as well (e.g., very low-dose scans with extreme noise), and suggest ways to improve the filter’s performance in these cases.

7. Finally, please provide a more thorough discussion of how fuzzy logic interacts with different noise types and how it compares to machine learning-based methods in terms of interpretability and robustness.

6. PLOS authors have the option to publish the peer review history of their article (what does this mean?). If published, this will include your full peer review and any attached files.

Reviewer #1: **Yes: **Abdel-Razzak Al-Hinnawi

Reviewer #2: No

---

## [Author Response · Author response to Decision Letter 0]

18 Nov 2024

Journal Requirements:

> Thank you for the templates, we have revised the style utilised.

> Thank you for the remainder, we have made the code available in the following repository: https://github.com/anicecloud/FGI_filter/. We will also update the data availability section of the submission to include it.

"This research has been supported by the TED project Grant Reference TED2021-131091B-I00 funded by MCIN/AEI/ 10.13039/501100011033 and by the "European Union NextGenerationEU/PRT''. Funding for open access charge: CRUE-Universitat Politècnica de València''

Please state what role the funders took in the study. If the funders had no role, please state:"The funders had no role in study design, data collection and analysis, decision to publish, or preparation of the manuscript''

> We need to amend the financial disclosure, since the funding for open access mentioned will not apply. The rest of it is correct, the funders had no role in study design, data collection and analysis, decision to publish, or preparation of the manuscript. The amended financial disclosure statement is:"This research has been supported by the TED project Grant Reference TED2021-131091B-I00 funded by MCIN/AEI/ 10.13039/501100011033 and by the "European Union NextGenerationEU/PRTR''. The funders had no role in study design, data collection and analysis, decision to publish, or preparation of the manuscript.''

"This research has been supported by the TED project Grant Reference TED2021-131091B-I00 funded by MCIN/AEI/ 10.13039/501100011033 and by the "European Union NextGenerationEU/PRTR'' '' 

" This research has been supported by the TED project Grant Reference TED2021-131091B-I00 funded by MCIN/AEI/ 10.13039/501100011033 and by the "European Union NextGenerationEU/PRTR''. Funding for open access charge: CRUE-Universitat Politècnica de València''

> Thank you, we will remove the statement from the Acknowledgements section.

Reviewer #1:

I- Moving the Material and metrics subsection to the Materials and method section (in the current text we have two same headings : 1-) Materials and Method 2-) Material and Metrics. why?

> Thank you for the comment, we agree with your observation that the subsection should be moved there, while the sections are not misnamed per se, we will rename the subsection for the sake of clarity. The subsection has been renamed "Dataset and metrics'' and moved to the "Materials and Methods'' section. Additionally, the header "Experiments'' has now been removed and its subsections are now directly under "Results''.

II- Adding a Table describing the number of CT volumes you experimented for each dataset (C095, L064, N005), or you mention them in the text clearly in the Material section.

> Each code corresponds to a different CT volume. We have rewritten the sentence so that it is more descriptive, from: 

"The volumes utilized correspond to CT scans of the chest (C095), abdomen (L064) and head (N005).''

to:

"The volumes utilized correspond to CT scans of the chest (volume C095 with 322 slices), abdomen (volume L064 with 209 slices) and head (volume N005 with 35 slices)''

III- Are there limitations to discuss with audience (or there are no limitations at all !!??) . Short paragraph in the discussion would be adequate.

> There exist hardware limitations, the performance results shown are valid only for the GPU utilized. The following paragraph has been added to the discussion:

"The Nsight analysis and the performance achieved in the results are valid for the characteristics of the machine utilized. There exist limitations when using the method in machines with different capabilities, especially with lower GPU memory, as the method is mostly memory bound. It could pose a problem in real-scenario applications, since hospital machines are not expected to have state of the art GPUs. However, these limitations affect computational speed but do not compromise image quality.''

IV- How you would expect the performance of the method if variable are changed (e.g. number of iterations, or computer hardware capabilities). Couple of sentences/statements in the discussion section would be adequate.

> Additionally to the previous paragraph, which addresses hardware capabilities, the following two paragraphs have been added to the discussion, addressing number of iterations and precision used:

"A limitation of the method is that previous to its implementation in a specific machine it would be necessary to carry out a heuristic study to determine the optimal number of iterations to maximize image quality. A variation in the number of iterations would also affect computational speed. However, a slight variation over the optimal would be in the range of milliseconds for computational speed and would not affect image quality significantly. 

Another aspect to consider when applying the method is precision. When testing a different precision, single instead of double, it was found that it did not have a noticeable impact in image quality and the impact in performance was not significant. Since the dataset DICOM data is originally stored in unsigned 16-bit integers, both single and double precision floating-point numbers are precise enough to represent the corresponding attenuation coefficients. Nevertheless, the kernels in single precision are still memory bound, so impact in performance is minimal. Additionally, double precision operations were used in the tests, if the filter was involved in the reconstruction process double precision would be expected, as precision has significant impact on reconstruction quality.''

V- Where is the table you mentioned in the last sentence in page 19 qualitative study subsection?

> Thank you for pointing it out, it disappeared due to an error with the latex formatting. It has been restored now.

VI- Finally, the reference list need some update work to be mentioned and discussed (how your method is different from them). Here are some examples that may be considerable of all of them or only the GPU ones at least :

https://journals.iucr.org/s/issues/2023/01/00/mo5264/mo5264.pdf

https://www.mdpi.com/1424-8220/24/6/1947

https://www.sciencedirect.com/science/article/pii/S0169260722001110

https://www.nature.com/articles/s41598-023-46028-9

https://www.sciencedirect.com/science/article/pii/S0010465523000644

https://dl.acm.org/doi/abs/10.1145/3650200.3656634

https://www.osti.gov/biblio/2429797

https://www.mdpi.com/2079-9292/10/24/3118

The above links are from Google Scholar published papers since 2020, you may search to find more in the topic. A paragraph or two in the discussion section would be adequate (i.e. comparing your method with the literature)

> Thank you for the provided references, we have included them in the following paragraphs:

"GPU-based acceleration techniques can be found in nearly all imaging modalities, and they provide faster computations than CPU, leading to real-time implementations, several techniques can be used to further optimize GPU-based applications. 

Pinned memory was used in this work to accelerate the load of data into GPU. Nikitin used this technique as well in the GPU-based tomographic reconstruction library in order to run data transfers concurrently to GPU computations. Lastly, Quintana et al. used it to accelerate data transfers between main memory and the GPU, as well as between disk and main memory. 

Further optimization can be achieved with the use of multiple GPUs. Heterogeneous GPUs were used by Chou et al. to resolve the memory-bound issue that appeared with iterative reconstruction at ultrahigh resolution in a single GPU. Similarly, Wu et al. utilize multi-GPUs to allow for the reconstruction of high-resolution CT volumes in real-time. Additionally, in that work the authors use several key features of modern GPUs, GPUDirect RDMA, GPUDirect Storage and Tensor Cores, which were used to further optimize backprojection when it is compute-bound.''

Reviewer #2:

1. Introduction section should include a more explicit hypothesis, such as: “This paper hypothesizes that a GPU-accelerated fuzzy filter can achieve both real-time processing and superior noise reduction in low-dose CT images compared to existing methods.”

> Thank you for the suggestion, we have added said phrase.

2. Please expand the literature review to include more recent developments in deep learning for medical image denoising, such as the use of convolutional neural networks (CNNs) and adversarial networks in CT image processing.

> We thank the reviewer for the suggestion. Accordingly, we have added the following text to the manuscript (please see the revised manuscript for the citations):

"In addition, new artificial intelligence-based filtering methods are emerging, applied to both general and medical imaging. In particular, for low-dose CT images, many methods have been presented in recent years, based on CNNs (Convolutional Neural Networks) and GANs (Generative Adversarial Networks). For instance, Li et al. propose a progressive cyclical convolutional neural network (PCCNN), a new unsupervised denoising framework using unpaired CT data. Patwari et al. propose a denoising framework which comprises two deep CNNs trained via a Deep-Q reinforcement learning task used to tune the parameters of the denoising methods used on both the CT images and the sinograms. As for GANs, Huang et al. propose DU-GAN, which leverages U-Net-based discriminators in the GAN framework to learn both global and local differences between the denoised and normal-dose images in both image and gradient domains. These are only a few examples of new AI-based denoising methods, but there are several extensive review papers on this issue.''

3. *While the technical description is thorough, the explanation of certain parameters (e.g., the fuzzy membership functions and the specific settings for CUDA optimization) could be more detailed. This is important for other researchers attempting to replicate the results.

> Thank you for your comment, the following sentences have been added or modified, in the case of parameter p_4:

"The p_3 utilized in the membership functions in [] was a function of the Gaussian noise applied in the simulated noise images (p_3 = (0.997 * sigma + 1.96)). Since we are using natural CT in attenuation coefficients, we have fixed the value as p_3 = 1.96/255.''

"The parameter p_4 was maintained as p_4 = 0.9 .''

4. My main concern about this research work is that the novelty is mainly technological, focusing on speedup and parallelization. A deeper discussion of how fuzzy logic enhances noise filtering in medical images, beyond computational efficiency, would strengthen the case for novelty.

> We thank the reviewer for the insight, we have added the following paragraph to the Materials and Methods section:

"The proposed fuzzy logic method is designed to handle mixed noise, specifically impulsive noise combined with either Gaussian or Poisson noise, by applying adaptive fuzzy rules and membership functions suited to each noise type’s characteristics. For Gaussian or Poisson noise (both of which typically produce a smoother, dispersed effect across the image) the fuzzy system employs similarity measures to weigh neighboring pixels. This approach averages out the background noise while preserving important structural details, ensuring that noise reduction does not blur critical edges. Impulsive noise, which appears as isolated, high-intensity pixels, is managed by calculating an impulsivity degree for each pixel; highly impulsive pixels receive lower weights in the filtering process, reducing their influence in the final image. By dynamically adjusting weights based on the unique attributes of Gaussian, Poisson, and impulsive noise, the fuzzy logic method achieves effective noise reduction in complex noise environments. This adaptability and interpretability make the method particularly valuable for medical imaging, where multiple noise types often coexist and where clarity of structural details is essential.''

5. While the method is evaluated on multiple datasets, the results could be strengthened by testing with additional, more diverse types of CT scans or by including additional performance metrics (e.g., time-to-result vs. hardware resource usage).

> Thank you for the suggestion, we have included time-to-result to the GPU executions on table 3, this is the time from program launch to finish, including load and store times.

6. Discuss in more detail the scenarios where the filter does not perform as well (e.g., very low-dose scans with extreme noise), and suggest ways to improve the filter’s performance in these cases.

> We thank the reviewer for the suggestion, the discussion was expanded with the following paragraph:

"Very low-dose scans present an amount of noise and artifacts that prove difficult to eliminate with the proposed filter. The recommendations for such high-noise is to eliminate noise before reconstruction, but when this is not possible there are other techniques that can be applied. Niwa et al. proposes an image-based forward projection and filtering of the artificial sinogram to reduce streak artifacts, while the work uses a Gaussian filter the FGI filter could be used instead. A different approach, that we will explore in future work, is the use of 3D windows with the idea of computing the areas obscured by artifacts by referencing the surrounding slices.''

7. Finally, please provide a more thorough discussion of how fuzzy logic interacts with different noise types and how it compares to machine learning-based methods in terms of interpretability and robustness.

> Thank you for your comment, we have added the following paragraph to the Results section:

"Through the use of membership functions and rule-based decision-making, the proposed fuzzy logic system can adjust seamlessly to varying noise levels, offering a customized approach for each noise type. This flexibility contrasts with machine learning methods, which typically require large datasets and extensive training to manage noise effectively. While machine learning models, particularly deep learning algorithms, can perform well under diverse noise conditions, they often function as opaque black boxes, limiting their interpretability. In contrast, the rule-based nature of fuzzy logic makes each filtering step traceable and clear, an essential quality in medical imaging wher

---

## [Decision Letter · Decision Letter 1]

11 Dec 2024

A Gpu-Accelerated Fuzzy Method for Real-Time CT Volume Filtering

PONE-D-24-40384R1

Dear Dr. Tendero Delicado,

We’re pleased to inform you that your manuscript has been judged scientifically suitable for publication and will be formally accepted for publication once it meets all outstanding technical requirements.

Kind regards,

Khan Bahadar Khan, Ph.D

Academic Editor

PLOS ONE

Additional Editor Comments (optional):

Reviewers' comments:

Reviewer's Responses to Questions

**Comments to the Author**

1. If the authors have adequately addressed your comments raised in a previous round of review and you feel that this manuscript is now acceptable for publication, you may indicate that here to bypass the “Comments to the Author” section, enter your conflict of interest statement in the “Confidential to Editor” section, and submit your "Accept" recommendation.

Reviewer #1: All comments have been addressed

Reviewer #2: All comments have been addressed

2. Is the manuscript technically sound, and do the data support the conclusions?

Reviewer #1: Yes

Reviewer #2: Yes

3. Has the statistical analysis been performed appropriately and rigorously? 

Reviewer #1: Yes

Reviewer #2: Yes

4. Have the authors made all data underlying the findings in their manuscript fully available?

Reviewer #1: Yes

Reviewer #2: Yes

5. Is the manuscript presented in an intelligible fashion and written in standard English?

Reviewer #1: Yes

Reviewer #2: Yes

6. Review Comments to the Author

Reviewer #1: (No Response)

Reviewer #2: Authors have successfully addressed my prior comments on the paper. Therefore, it is now ready for publication.

7. PLOS authors have the option to publish the peer review history of their article (what does this mean?). If published, this will include your full peer review and any attached files.

Reviewer #1: **Yes: **Abdel-Razzak Al-Hinnawi

Reviewer #2: No

---

## [Editor Report · Acceptance letter]

18 Dec 2024

PONE-D-24-40384R1 

PLOS ONE

Dear Dr. Tendero Delicado, 

I'm pleased to inform you that your manuscript has been deemed suitable for publication in PLOS ONE. Congratulations! Your manuscript is now being handed over to our production team.

Kind regards, 

on behalf of

Dr. Khan Bahadar Khan 

Academic Editor

PLOS ONE